# InfoPrune: Revisiting Visual Token Pruning from an Information-Theoretic Perspective

## Abstract

Multimodal large language models (MLLMs) rely on dense visual tokens, but their indiscriminate propagation causes severe inference overhead. Existing pruning strategies largely treat token importance as a static property (e.g., attention strength), overlooking the dynamic nature of evidence flow. In this work, we re-cast pruning as an information maximization problem under budget constraints: under limited computation, which tokens provide genuine marginal information, and when has their contribution been fully injected into the language stream? Guided by this formulation, Guided by this formulation, we propose **InfoPrune**, a training-free two-stage framework. Stage 1 refines visual token selection by combining attention priors with information increment, while Stage 2 detects mid-layer semantic convergence and performs one-shot pruning within the LLM. This design directly targets "who to keep" and "when to stop," reducing redundancy while preserving essential semantics. Experiments on LLaVA-1.5, LLaVA-Next, and Qwen-VL-2.5 show that InfoPrune achieves over 96% performance retention with only 11.1% tokens, outperforming prior methods in generality, stability, and efficiency. Our work provides both a principled perspective on multimodal evidence budgeting and a practical solution for efficient inference.

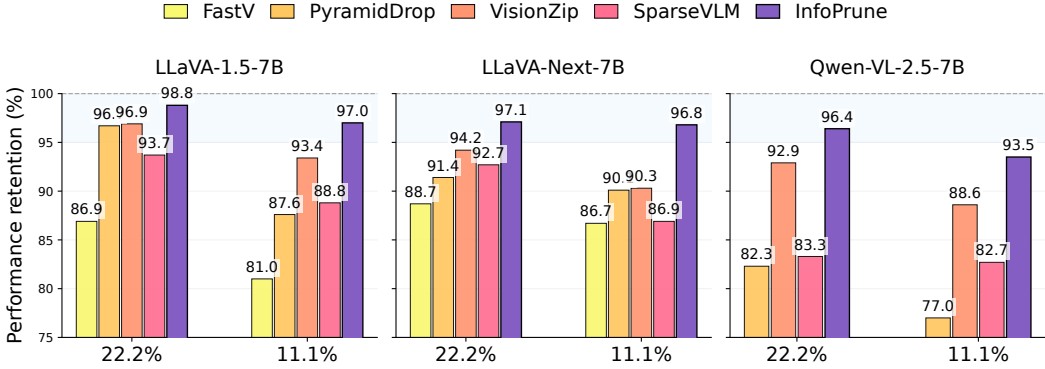

Figure 1: **The performance of InfoPrune.** The x-axis denotes the proportion of visual tokens retained. InfoPrune achieves substantial performance preservation across multiple models compared to several baselines.

## 1 Introduction

Multimodal large language models (MLLMs) have achieved remarkable progress in tasks such as visual question answering, image-text reasoning, and caption generation. By encoding images as sequences of visual tokens and integrating them with language tokens, MLLMs substantially improve multimodal understanding. However, their impressive capabilities come at a high computational cost. Visual tokens are typically propagated through all layers without selection, and their volume and depth of interaction result in significantly increased inference latency, memory consumption, and FLOPs—posing a major challenge for real-world deployment.

To alleviate this computational bottleneck, researchers have proposed *visual token pruning* strategies that aim to reduce overhead by discarding redundant visual tokens during inference. Existing methods can be broadly categorized into two groups: the first performs importance-based selection in the visual encoder using attention or saliency scores (Yang et al., 2024) (Vasu et al., 2025), while the second prunes tokens (Chen et al., 2024) (Yin et al., 2025) (Xing et al., 2025) inside the LLMs to prevent deep-layer propagation. However, these methods often rely on a simplified assumption: that token importance can be directly inferred from local features or static scores (e.g., attention weights).

While such heuristics have shown empirical success, their core premise—that important tokens are easily identifiable via static features—fails to capture the true dynamics of token-level information flow within the model. Even in models with strong overall performance, we observe that tokens selected by attention-based pruning may still provide insufficient evidence, ultimately leading to reasoning failures. This highlights a more fundamental oversight: the model does not necessarily need tokens that merely appear important, but rather those that actively contribute to the semantic transmission process.

In this work, we propose a new perspective: *visual tokens are not static inputs to be processed, but dynamic evidence carriers injected into the language model across layers.* Based on this standpoint, we frame visual token pruning as an information maximization problem under budget constraints:

> **Problem**
>
> *Under constrained computation, how can we optimally preserve tokens that bring high marginal information increments, and terminate their involvement once their information has been fully injected?*

This core problem can be decomposed into two subproblems that reflect two distinct but complementary pruning perspectives:

> **Subproblem I**
>
> *Visual-centric pruning: Which visual tokens contribute truly new information?*

> **Subproblem II**
>
> *Text-centric pruning: When should visual token injection be halted to avoid computation that is costly yet semantically redundant?*

To systematically address these subproblems, we adopt an **information-theoretic lens** and conduct in-depth empirical and theoretical analyses across mainstream MLLMs. From these analyses, we derive two key insights: *(1)While attention scores offer a useful prior for token relevance, they mainly capture local saliency and do not reflect whether a token contributes new information. This suggests that marginal information gain should be explicitly measured, so that token selection favors truly informative and non-redundant inputs. (2) By the middle layers, visual information has been sufficiently injected, and the language semantics have stabilized. Further retaining all visual tokens beyond this point yields diminishing returns and introduces redundant computation.*

Grounded in these insights, we propose **InfoPrune**, a two-stage **info**rmation-theoretic **prun**ing framework that leverages attention priors and information increment to guide token selection, and dynamically halts visual token propagation once their contribution saturates. This design reduces redundancy while preserving critical visual semantics.

Extensive experiments on LLaVA-1.5, LLaVA-Next, and Qwen-VL-2.5 demonstrate that InfoPrune achieves strong performance retention under aggressive token reduction. As shown in Fig 1, **LLaVA-1.5** retains **98.8%** and **97.0%** average performance when preserving only **22.2%** and **11.1%** visual tokens, respectively. **LLaVA-Next** achieves **97.1%** and **96.8%** under the same pruning ratios, while **Qwen-VL-2.5** maintains **96.4%** and **93.5%**. These results highlight InfoPrune's superior generalizability, stability, and information efficiency over existing approaches.

Our main contributions are as follows:

- We recast visual token pruning in MLLMs as an *information maximization problem under budget constraints*, grounded in the flow of evidential signals. This formulation naturally

decomposes into two subproblems: identifying tokens with high marginal information gain, and determining when visual evidence injection is complete.

- We propose InfoPrune, a training-free two-stage pruning framework. The first stage combines attention priors with information increment to prune tokens. The second stage detects semantic stabilization in the language model and performs one-shot mid-layer pruning, significantly improving inference efficiency.

- We conduct comprehensive experiments on three mainstream MLLMs: LLaVA-1.5, LLaVA-Next, and Qwen-VL-2.5. Results show that InfoPrune consistently outperforms prior state-of-the-art methods, achieving over **96.8%** performance retention under extreme compression (e.g., retaining only **11.1%** of visual tokens), while exhibiting strong generalization and robustness.

## 2 MOTIVATION

To operationalize our formulation of visual token pruning as an information planning problem, we now turn to empirical and theoretical analyses of the two subproblems introduced in Section 1. Specifically, we ask: (1) how effective are current token selection strategies at identifying marginally informative tokens, and (2) at what point in the model has visual information been sufficiently absorbed to warrant pruning? These analyses guide the design of our proposed method.

### 2.1 VISION-CENTRIC PRUNING: AN INFORMATION-INCREMENTAL PERSPECTIVE

As the core of visual-centric pruning, Subproblem I asks: *Given partial evidence already retained, how can we identify visual tokens that still contribute meaningful new information?* Intuitively, such tokens should carry semantic signals that cannot be linearly expressed by the current token subset, thereby serving as marginally informative additions to the visual context. From an information-theoretic perspective, the selection process should exhibit marginal optimality—each retained token should significantly expand the representational subspace. While this goal is inherently structural, most existing approaches approximate token importance via attention scores assigned by the [CLS] token, under the assumption that higher attention implies higher utility. Despite their empirical success and interpretability, we find this assumption often fails in practice. The tokens selected by top attention scores may collectively fail to support accurate reasoning, suggesting that attention strength alone is an unreliable proxy for informational value.

To more faithfully capture token utility, we propose to model the selection process as a subspace expansion problem, where each candidate token is evaluated by its ability to introduce a new direction orthogonal to the span of previously selected tokens. This residual signal—defined as the orthogonal component of a token with respect to the current subspace—serves as a measure of its marginal information increment. In the setting of token-budget constrained pruning, we thus formulate token selection as a subset optimization problem: given a budget $T$, select a subset $\mathcal{S} \subseteq \{1, \ldots, N\}$ such that the selected tokens preserve as much global semantic information as possible. To support this formulation with tractable and theoretically grounded objectives, we introduce two complementary first-order indicators that capture each token's local novelty and global structural contribution, which we elaborate in the next section.

First, if information sufficiency is understood as covering the principal directions of the full set, the natural objective is to maximize the geometric volume of the selected vectors, *i.e.*, the Gram determinant $\log \det G_{\mathcal{S}}$. In incremental selection, the marginal gain of adding a candidate $x_i$ can be expressed by the Schur complement as

$$\log \det G_{\mathcal{S} \cup \{i\}} - \log \det G_{\mathcal{S}} \;=\; \log\big(1 + \|r_i\|_2^2\big), \tag{1}$$

where $r_i = x_i - \mathbf{U}\mathbf{U}^\top x_i$ is the orthogonal residual of $x_i$ with respect to the subspace spanned by $X_{\mathcal{S}}$, and $\mathbf{U}$ is its orthonormal basis. Thus, the marginal volume growth is monotonically aligned with the residual norm, which motivates our first indicator - Marginal Feature Increment (MFI):

$$\mathrm{MFI}(x_i \mid \mathcal{S}) = \|x_i - \mathbf{U}\mathbf{U}^\top x_i\|_2, \tag{2}$$

measuring the *non-reconstructability* or new directionality of a candidate relative to the current subspace. Since $\log \det G_{\mathcal{S}}$ is submodular, this residual-based greedy selection enjoys natural approximation guarantees.

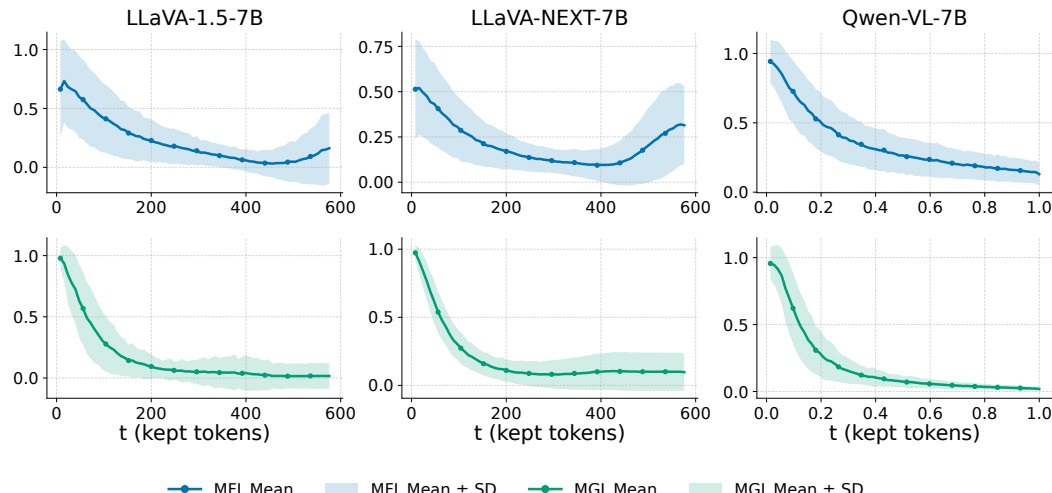

Figure 2: **Visualization of Marginal Feature Increment (MFI) & Marginal Gain of Information (MGI) under the attention-based token selection strategy on the MME dataset with LLaVA-1.5-7B, LLaVA-Next and Qwen-VL-2.5.** All curves are computed over multiple samples, with shaded areas indicating standard deviations.

Second, if information sufficiency is viewed in terms of maximizing parameter identifiability under a linear–Gaussian proxy (*i.e.*, Fisher information or Gaussian entropy), the objective becomes the D-optimal design $\max_{|\mathcal{S}|=T} \log \det(C_\mathcal{S} + \varepsilon I)$, where $C_\mathcal{S} = \frac{1}{|\mathcal{S}|} \sum_{j \in \mathcal{S}} x_j x_j^\top$. In the incremental view, the marginal statistical gain of adding $x_i$ is Marginal Gain of Information (MGI):

$$\text{MGI}(x_i \mid \mathcal{S}) = \log \det(C_{\mathcal{S} \cup \{i\}} + \varepsilon I) - \log \det(C_\mathcal{S} + \varepsilon I), \tag{3}$$

which quantifies the improvement in covariance volume (statistical diversity) brought by the candidate, *i.e.*, its *global structural contribution*. Owing to the submodularity of $\log \det$, this criterion naturally favors complementary directions while penalizing redundancy.

Under an information-optimal selection order, MFI should *decay rapidly toward zero* as steps proceed, while the marginal gain of MGI should *gradually diminish toward zero*. Significant oscillations or non-convergent patterns, in contrast, typically signal redundancy or failure to introduce effective new directions. Further analysis of these two metrics can be found in C.2.2.

After constructing the MFI and MGI metrics, we evaluated the performance of several mainstream models under the standard attention-based Top-$k$ token selection strategy. Each token selection step is treated as a time index $t$, and we plot the corresponding MFI and MGI distributions across tokens.

Fig 2 shows that the selected tokens under this strategy exhibit considerable variance across samples in both metrics, indicating a lack of consistency. More notably, the MFI curves reveal a clear "late-stage advantage": tokens ranked lower in attention often yield higher marginal information gains than earlier ones. This suggests a fundamental mismatch—high attention scores do not necessarily imply high information contribution.

Based on the above analysis, we arrive at the core insight I for Subproblem I:

> **Core Insight I**
>
> *(1)While attention scores offer a useful prior for token relevance, they mainly capture local saliency and do not reflect whether a token contributes new information. This suggests that marginal information gain should be explicitly measured, so that token selection favors truly informative and non-redundant inputs.*

## 2.2 TEXT-CENTRIC PRUNING: PERSPECTIVE OF INFORMATION INJECTION

Text-centric pruning addresses the second core question in our evidence budgeting framework:

**Which visual tokens have already fulfilled their role in information transmission and can thus be safely removed?**

To analyze this, we adopt two complementary perspectives:

- **Visual-to-textual information flow.** We track how much visual evidence is injected into the text stream across layers, identifying when further token retention yields diminishing returns.
- **Stability of semantic predictions.** We monitor the convergence of next-token predictions to detect when the language model has fully internalized visual information.

To characterize how visual tokens propagate and inject information into the language model, we design two layer-wise indicators: (1) the *visual-to-textual flow* $S_{vt}(\ell)$, measuring the information injected from visual tokens to text tokens at each layer $\ell$; and (2) the *intra-visual flow* $S_{vv}(\ell)$, quantifying the interaction strength among visual tokens themselves.

Instead of relying on raw attention weights, we adopt a saliency-based measure that captures token-to-token information flow. Specifically, the flow from token $j$ to token $i$ at layer $\ell$ is defined as:

$$I_\ell(i,j) = \left\| \frac{\partial \mathcal{L}}{\partial h_i^L} \cdot \mathbf{W}_\ell^V \cdot A_\ell(i,j) \right\|_2 \tag{4}$$

where $\mathcal{L}$ is the model's loss function, $h_i^L$ denotes the final-layer representation of token $i$, $\mathbf{W}_\ell^V$ is the value projection matrix, and $A_\ell(i,j)$ is the attention from token $i$ to token $j$. The saliency score $I_\ell(i,j)$ reflects the contribution of token $j$ to the loss-relevant representation of token $i$. Based on this formulation, we define the two indicators as:

$$S_{vt}(\ell) = \frac{1}{|V|} \sum_{j \in I} \sum_{i \in V} I_\ell(i,j), \quad S_{vv}(\ell) = \frac{1}{|V|} \sum_{j \in V} \sum_{i \in V} I_\ell(i,j) \tag{5}$$

where $\mathcal{V}$ and $\mathcal{I}$ denote the sets of visual and text tokens, respectively. These two indicators quantify the *injectivity* and *redundancy* of visual tokens at each layer, offering structural guidance for deciding **when** to stop propagating them.

As shown in Fig 3, model exhibit a consistent trend: the strength of visual-to-text injection begins to decline noticeably from the middle layers, and the overall degree of cross-modal interaction diminishes as depth increases. This suggests that the language model has already absorbed sufficient visual information by the mid-stage, providing empirical support for pruning at a saturation point. See C.1 for more results.

Beyond quantifying visual injection, we assess whether the language model has reached a state of semantic maturity—*i.e.*, whether the internal representation has stabilized sufficiently to support next-token prediction. Instead of comparing embeddings across layers, we adopt a behavioral view by tracking the model's predictive dynamics. At each transformer layer, we extract the

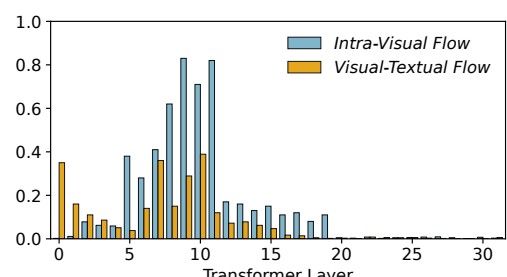

Figure 3: **Layer-wise evolution of $S_{vt}$ and $S_{vv}$ for LLaVA-1.5 on POPE.** Visual information is massively injected from shallow to mid layers, after which its intensity steadily decays.

next-token distribution and monitor two signals: (1) changes in the top-1 predicted token, and (2) the stability of its predicted probability.

As shown in Fig 15, early layers exhibit frequent shifts in both the predicted token and its confidence, indicating semantic volatility. In contrast, middle layers mark a convergence point: the top-1 token stabilizes, and its probability becomes increasingly consistent. This behavioral plateau persists in deeper layers, suggesting that the model has effectively consolidated the input semantics.

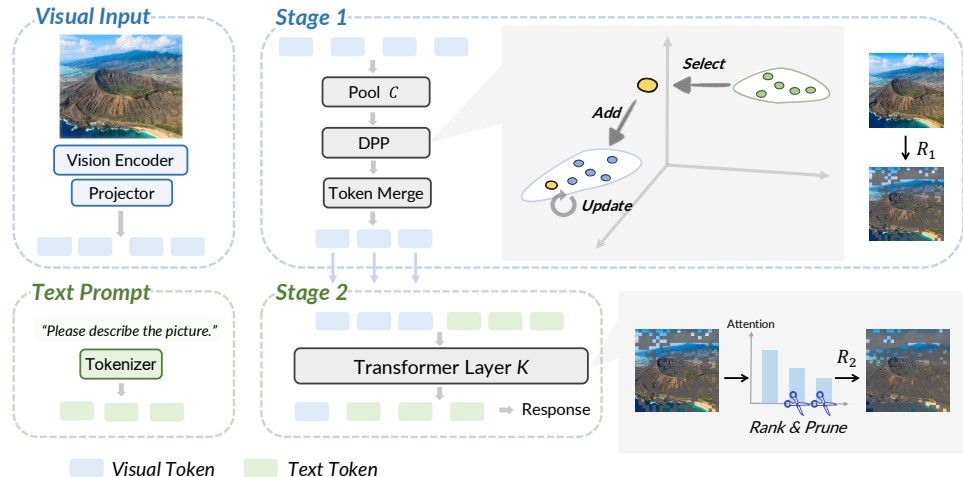

Figure 4: **Overview of the InfoPrune framework.** Stage 1 performs vision-side pruning: starting from $N$ visual tokens, we first build a candidate pool $C$ with attention priors, then apply a Determinantal Point Process (DPP) selection focusing on marginal information gain. After obtaining the retained tokens, a token merge step absorbs the information from discarded tokens, resulting in $T_1 = R_1 \cdot N$ informative tokens. Stage 2 performs LLM-side pruning: once semantic convergence is detected at a mid-layer $K$, a further ratio $R_2$ is applied to prune the $T_1$ tokens in one shot, leaving $T_2 = R_2 \cdot T_1$ tokens. In the DPP process, we first *select* the token with the largest residual (yellow) from the unselected candidates (green) and then *add* it to the selected set $\mathcal{S}$ (blue); the orthogonal basis $\mathbf{U}$ of $\mathcal{S}$ is subsequently *updated* to reflect the token addition.

Interestingly, this trend aligns with the saturation point of visual-to-textual injection, reinforcing our hypothesis that visual information has been fully consumed by the language model by mid-depth.

Based on the above analysis, we arrive at the core insight II for Subproblem II:

> **Core Insight II**
>
> *By the middle layers, visual information has been sufficiently injected, and the language semantics have stabilized. Further retaining all visual tokens beyond this point yields diminishing returns and introduces redundant computation.*

## 3 METHOD

Building on the two key insights, we introduce InfoPrune, an information-driven, two-stage pruning framework, as shown in Fig 4.

In the first stage, InfoPrune prunes on the vision encoder side by constructing an attention-guided candidate pool and selecting a diverse subset based on marginal information gain. In the second stage, it prunes inside the LLM backbone by detecting semantic saturation at an intermediate layer and performing text token removal, substantially reducing downstream computation with minimal loss of information. We now elaborate on the design and implementation of each stage.

### 3.1 STAGE 1: VISUAL ENCODER PRUNING AND MERGE

The first stage of **InfoPrune** performs token pruning within the visual encoder. The objective is to compress the original set of $N$ visual tokens down to $T_1 \ll N$ tokens, while preserving sufficient semantic content for downstream reasoning. We define the visual-side pruning ratio as $R_1 = T_1/N$. This stage comprises three steps: attention-based candidate pool $C$, Determinantal Point Process (DPP) algorithm based on marginal information gain, and token merge. The full procedure is shown in Algo 1.

We begin by extracting deep attention scores $a_i$ from $\texttt{[CLS]}$ token within the vision encoder. These scores reflect the global saliency of each visual token. We apply a softmax normalization with temperature $\tau_{\text{attn}}$, followed by a sharpening exponent $\gamma_q$, to obtain the attention prior:

$$q_i = \left( \text{softmax} \left( \frac{a}{\tau_{\text{attn}}} \right)_i \right)^{\gamma_q}, \quad i = 1, \ldots, N \tag{6}$$

**Attention-based candidate pool $C$:** We then select the top-$\alpha T$ tokens with highest $q_i$ values to form a candidate pool $\mathcal{C}$, where $\alpha > 1$ is an over-provisioning factor.

**DPP-algorithm:** For each candidate, we construct a DPP-style feature:

$$Z_i = \sqrt{q_i} \cdot x_i, \quad i \in \mathcal{C} \tag{7}$$

From this pool, we perform greedy selection using a residual-based marginal information gain strategy (Kulesza & Taskar, 2012). Let $\mathbf{U} \in \mathbb{R}^{d \times |S|}$ denote the current orthonormal basis of the subspace spanned by the selected tokens $\mathcal{S}$. For each candidate $Z_i$, we compute its squared residual:

$$r_i = \|Z_i\|_2^2 - \|\mathbf{U}^\top Z_i\|_2^2 = \|(I - P_U)Z_i\|_2^2 \tag{8}$$

The token with the largest $r_i$ is added to $\mathcal{S}$, and $\mathbf{U}$ is updated accordingly (See Alg 1 for details). The process stops early if the maximum residual falls below a threshold $\varepsilon$, indicating novelty saturation. If $|S| < T_1$, we fill the remaining slots using the top-$q_i$ tokens from the leftover candidates; if $|S| > T_1$, we truncate $\mathcal{S}$ using $q_i$ scores to satisfy the budget $|S| = T_1$.

**Token Merge:** To preserve tensor shape while retaining semantic cues from discarded tokens, we perform prototype-based feature consolidation. Each dropped token is softly merged into its most similar retained token using cosine similarity and inertia-weighted accumulation. The fused features are written back only at the selected positions, resulting in an output tensor $\tilde{\mathbf{X}} \in \mathbb{R}^{N \times d}$ compatible with downstream processing. See Alg 1 for details.

### 3.2 Stage 2: Mid-layer One-shot Pruning in the LLM

The second stage of InfoPrune performs a one-shot pruning at a predefined intermediate layer $K$ in the LLM to remove redundant visual tokens after their semantic information has been largely absorbed.

Given $T_1$ tokens from Stage 1 and a target pruning ratio $R_2 \in (0, 1]$, we retain $T_2 = R_2 \cdot T_1$ tokens by ranking their importance based on attention from the final text token at layer $K$. Specifically, we compute:

$$s_i = \frac{1}{H} \sum_{h=1}^{H} \alpha_i^{(K,h)}, \quad i = 1, \ldots, T_1 \tag{9}$$

The top-$T_2$ tokens by score are preserved, while others are removed. Both hidden states and attention masks are updated, and only the selected tokens are propagated beyond layer $K$.

## 4 Results

We deploy our proposed method on a range of widely used multimodal models and conduct systematic comparisons against several representative baselines across multiple popular benchmarks.

### 4.1 Settings

#### 4.1.1 Models & Baseline & Benchmark

**Models:** We evaluate our method on three representative and widely adopted open-source vision-language models: LLaVA-1.5, LLaVA-NeXT, and Qwen-VL-2.5. These models have become standard backbones in the community and serve as the foundation for many subsequent MLLM developments.

**Baselines:** We compare against a set of strong and community-recognized baselines, including FastV (Chen et al., 2024), PyramidDrop (Xing et al., 2025), VisionZip (Yang et al., 2024), and

Table 1: **Performance comparisons on LLaVA-1.5-7B across 9 benchmarks.** The best results in each setting are **bolded**.

| Method | Venue | GQA | MMB | MMB$^{CN}$ | MME | POPE | SQA$_{IMG}$ | VizWiz | VQAText | VQAv2 | Average |
|---|---|---|---|---|---|---|---|---|---|---|---|
| *Upper Bound, 576 Tokens (100%)* | | | | | | | | | | | |
| LLaVA-1.5-7B (Liu et al., 2023a) | *NeurIPS'23* | 61.9 | 64.7 | 58.1 | 1862 | 85.9 | 69.5 | 50.0 | 58.2 | 78.5 | 100.0% |
| *Retain 192 Tokens in Average (↓66.7%)* | | | | | | | | | | | |
| FastV (Chen et al., 2024) | *ECCV'24* | 53.4 | 60.9 | 56.1 | 1658 | 65.1 | 67.3 | 50.3 | 54.7 | 67.1 | 91.0% |
| PyramidDrop (Xing et al., 2025) | *CVPR'25* | 57.1 | 63.3 | 56.0 | 1791 | 83.1 | 68.9 | **51.0** | 56.8 | 75.4 | 97.1% |
| VisionZip (Yang et al., 2024) | *CVPR'25* | 59.3 | 63.0 | 57.1 | 1783 | 85.3 | 68.0 | 50.3 | 57.3 | 74.7 | 96.7% |
| SparseVLM (Zhang et al., 2025) | *ICML'25* | 57.6 | 62.5 | 53.7 | 1721 | 83.6 | **69.1** | 50.5 | 56.1 | 75.6 | 96.1% |
| **Ours** | - | **60.5** | **64.0** | **57.5** | **1817** | **86.2** | 68.6 | 50.8 | **58.1** | **77.7** | **99.2%** |
| *Retain 128 Tokens in Average (↓77.8%)* | | | | | | | | | | | |
| FastV (Chen et al., 2024) | *ECCV'24* | 51.5 | 56.1 | 55.9 | 1534 | 59.9 | 62.1 | 51.3 | 52.6 | 64.3 | 86.9% |
| PyramidDrop (Xing et al., 2025) | *CVPR'25* | 57.7 | 61.0 | 56.6 | 1765 | 82.3 | 69.0 | 50.8 | 56.7 | 75.6 | 96.7% |
| VisionZip (Yang et al., 2024) | *CVPR'25* | 56.8 | 62.0 | 56.6 | 1763 | 83.2 | 68.9 | 51.3 | 56.8 | 75.6 | 96.9% |
| SparseVLM (Zhang et al., 2025) | *ICML'25* | 56.0 | 60.0 | 51.1 | 1696 | 80.5 | 67.1 | 51.4 | 54.9 | 73.8 | 93.7% |
| **Ours** | - | **59.7** | **63.2** | **57.8** | **1780** | **86.1** | **69.6** | **51.6** | **57.0** | **77.0** | **98.8%** |
| *Retain 64 Tokens in Average (↓88.9%)* | | | | | | | | | | | |
| FastV (Chen et al., 2024) | *ECCV'24* | 50.1 | 54.7 | 53.0 | 1329 | 48.6 | 55.1 | 50.9 | 49.1 | 61.6 | 81.0% |
| PyramidDrop (Xing et al., 2025) | *CVPR'25* | 52.5 | 58.0 | 50.5 | 1569 | 55.9 | 69.2 | 50.7 | 50.5 | 70.3 | 87.6% |
| VisionZip (Yang et al., 2024) | *CVPR'25* | 55.1 | 61.0 | 54.9 | 1690 | 77.0 | 63.0 | 52.1 | 55.5 | 72.4 | 93.4% |
| SparseVLM (Zhang et al., 2025) | *ICML'25* | 55.2 | 56.0 | 50.4 | 1532 | 75.1 | 62.1 | 49.2 | 51.0 | 71.8 | 88.8% |
| **Ours** | - | **57.3** | **61.8** | **56.0** | **1715** | **85.0** | **69.8** | **52.6** | **55.9** | **75.0** | **97.0%** |

Table 2: **Performance comparisons on LLaVA-NEXT-7B across 9 image understanding benchmarks.** The best results in each setting are **bolded**.

| Method | Venue | GQA | MMB | MMB$^{CN}$ | MME | POPE | SQA$_{IMG}$ | VizWiz | VQAText | VQAv2 | Average |
|---|---|---|---|---|---|---|---|---|---|---|---|
| *Upper Bound, 2880 Tokens (100%), 3.817 TFLOPs* | | | | | | | | | | | |
| LLaVA-NEXT-7B (Li et al., 2024) | *arXiv'24* | 64.2 | 67.4 | 60.6 | 1851 | 86.5 | 70.1 | 57.6 | 61.3 | 81.8 | 100.0% |
| *Retain 22.2% Tokens in Average (↓77.8%)* | | | | | | | | | | | |
| FastV (Chen et al., 2024) | *ECCV'24* | 55.9 | 61.6 | 51.9 | 1661 | 71.7 | 62.8 | 53.1 | 55.7 | 72.9 | 88.7% |
| PyramidDrop (Xing et al., 2025) | *CVPR'25* | 56.4 | 63.4 | 56.2 | 1663 | 77.6 | 67.5 | 54.1 | 54.4 | 73.5 | 91.4% |
| VisionZip (Yang et al., 2024) | *CVPR'25* | 59.3 | 63.1 | 58.7 | 1702 | 82.1 | 67.3 | 53.3 | 58.9 | 76.2 | 94.2% |
| SparseVLM (Zhang et al., 2025) | *ICML'25* | 57.7 | 64.3 | 57.3 | 1694 | 81.7 | 67.3 | 52.5 | 55.9 | 73.4 | 92.7% |
| **Ours** | - | **62.4** | **68.0** | **60.5** | **1797** | **84.5** | **68.9** | **53.1** | **57.2** | **79.4** | **97.1%** |
| *Retain 11.1% Tokens in Average (↓88.9%)* | | | | | | | | | | | |
| FastV (Chen et al., 2024) | *ECCV'24* | 52.7 | 59.8 | 51.1 | 1577 | 69.1 | 68.4 | 51.3 | 53.3 | 71.1 | 86.7% |
| PyramidDrop (Xing et al., 2025) | *CVPR'25* | 54.8 | 61.9 | 54.7 | 1619 | 75.9 | 68.0 | 53.8 | 54.1 | 72.9 | 90.1% |
| VisionZip (Yang et al., 2024) | *CVPR'25* | 55.5 | 60.1 | 56.1 | 1630 | 74.8 | 68.3 | 54.0 | 56.2 | 71.4 | 90.3% |
| SparseVLM (Zhang et al., 2025) | *ICML'25* | 51.2 | 63.1 | 55.1 | 1542 | 76.3 | 67.5 | 53.8 | 46.4 | 66.3 | 86.9% |
| **Ours** | - | **61.5** | **67.7** | **60.0** | **1797** | **85.0** | **68.7** | **53.7** | **57.1** | **78.4** | **96.8%** |

SPARSEVLM (Zhang et al., 2025). These methods cover a diverse range of pruning strategies,, such as attention pruning, token dropping, and multi-scale evidence selection.

**Benchmark:** We evaluate our method on 9 benchmarks, including GQA (Hudson & Manning, 2018), MMB (Liu et al., 2023b), MMB$^{CN}$ (Li et al., 2022), MME (Fu et al., 2023), POPE (Li et al., 2023), SQA$_{IMG}$ (Huang et al., 2023), VizWiz (Gurari et al., 2018), VQAText (Singh et al., 2019), and VQAv2 (Goyal et al., 2016).

### 4.1.2 DETAILS

Please see Appendix B for implementation details.

### 4.2 MAIN RESULTS

**Results in LLaVA-1.5-7B:** As shown in Tab 1, InfoPrune achieves the best performance across all compression rates, retaining **99.2%**, **98.8%**, and **97.0%** accuracy under 192, 128, and 64 tokens, respectively. This consistently surpasses strong baselines such as VisionZip (96.7%), PyramidDrop (96.7%), and SparseVLM (96.1%), confirming its robustness and generalization under tight budgets. Notably, InfoPrune maintains top scores on critical reasoning benchmarks (e.g., GQA, POPE, and VQAv2), while preserving competitive accuracy even on challenging datasets, demonstrating its effectiveness in retaining essential visual evidence for multi-modal reasoning.

**Results in LLaVA-NEXT-7B:** As shown in Tab 2, InfoPrune achieves state-of-the-art performance under both moderate and aggressive pruning, retaining **97.1%** accuracy with 320 tokens (77.8% reduction) and **96.8%** with only 160 tokens (88.9% reduction). These results substantially surpass prior methods such as VisionZip (94.2%) and PyramidDrop (91.4%), while competing base-

Table 3: **Performance comparisons on Qwen-VL-2.5-7B across 6 benchmarks.** The best results in each setting are **bolded**.

| Method | Venue | MMB | MMB$^{CN}$ | MME | POPE | GQA | VQAv2 | Average |
|---|---|---|---|---|---|---|---|---|
| *Upper Bound, 100%* | | | | | | | | |
| Qwen-VL-2.5-7B  (Bai et al., 2025) | *arXiv'25* | 83.2 | 82.8 | 2347 | 86.7 | 61.6 | 83.9 | 100.0% |
| *Retain 22.2% Tokens on Average (↓77.8%)* | | | | | | | | |
| PyramidDrop  (Xing et al., 2025) | *CVPR'25* | 59.6 | 68.6 | 2076 | 74.6 | 51.1 | 68.7 | 82.3% |
| VisionZip  (Yang et al., 2024) | *CVPR'25* | 77.8 | 75.2 | 2151 | 84.7 | 56.7 | 76.9 | 92.9% |
| SparseVLM  (Zhang et al., 2025) | *ICML'25* | 64.3 | 57.3 | 1694 | 81.7 | 57.8 | 78.2 | 83.3% |
| **Ours** | - | **78.6** | **76.9** | **2131** | **87.5** | **62.3** | **82.5** | **96.4%** |
| *Retain 11.1% Tokens on Average (↓88.9%)* | | | | | | | | |
| PyramidDrop  (Xing et al., 2025) | *CVPR'25* | 56.1 | 65.7 | 1914 | 71.2 | 45.6 | 65.2 | 77.0% |
| VisionZip  (Yang et al., 2024) | *CVPR'25* | 74.5 | 73.1 | 1991 | 82.8 | 55.3 | 70.4 | 88.6% |
| SparseVLM  (Zhang et al., 2025) | *ICML'25* | 63.1 | 55.1 | 1951 | 76.3 | 57.4 | 75.3 | 82.7% |
| **Ours** | - | **76.2** | **74.4** | **2004** | **85.0** | **62.1** | **79.9** | **93.5%** |

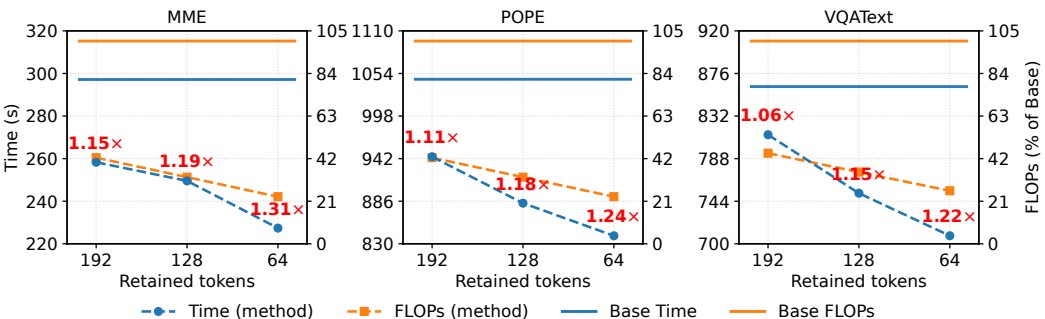

Figure 5: **Effectiveness of InfoPrune on LLaVA-1.5 across three benchmarks.**

lines drop below 87%. Despite minor fluctuations on VizWiz and VQAText, InfoPrune consistently ranks top across benchmarks, demonstrating strong robustness and efficiency trade-offs in large-scale models.

**Results in Qwen-VL-2.5:** InfoPrune shows clear advantages under both moderate and aggressive pruning, retaining **96.4%** performance with 22.2% tokens and **93.5%** with only 11.1% tokens, while baselines drop below 89%. This demonstrates its superior accuracy–efficiency trade-off and robustness under extreme compression.

## 4.3 EFFICIENCY STUDY

**Efficiency Study:** As shown in Fig 5, reducing the number of retained tokens on **LLaVA-1.5-7B** consistently decreases inference cost. FLOPs shrink to about 23% under aggressive pruning, yielding up to **1.5×** speedup across MME, POPE, and VQAText, while maintaining strong accuracy. These results confirm that InfoPrune offers substantial efficiency gains with minimal performance loss.

## 5 CONCLUSION

We revisit visual token pruning in MLLMs from an information-theoretic perspective, formulating two key subproblems: which tokens provide high marginal information, and when visual injection should stop. We propose **InfoPrune**, a two-stage framework that combines attention priors with information increment for token selection, and detects semantic saturation to enable one-shot pruning in the LLM. Experiments on LLaVA-1.5, LLaVA-Next, and Qwen-VL-2.5 show that InfoPrune substantially reduces computation while preserving accuracy, consistently outperforming prior methods and establishing a principled paradigm for efficient multimodal reasoning.

## ETHICS STATEMENT

This work focuses on methodological contributions for improving the efficiency of MLLMs through information-theoretic token pruning. Our study does not involve human subjects, personal data, or sensitive demographic attributes. All experiments are conducted on widely used, publicly available benchmark datasets (e.g., GQA, VQAv2, MMBench), which are released under academic or research-friendly licenses.

By substantially reducing inference cost without sacrificing accuracy, our method has the potential to lower the environmental and financial footprint of large-scale multimodal model deployment, thereby contributing positively to the accessibility and sustainability of AI research. Nevertheless, as with any efficiency-oriented technique, there remains the possibility that accelerated inference could facilitate misuse (e.g., in generating harmful or biased content more efficiently). We emphasize that such risks stem from downstream usage rather than from our pruning framework itself, and we encourage responsible application aligned with the ICLR Code of Ethics.

We affirm that this work complies with ethical standards, respects dataset usage guidelines, and raises no conflicts of interest or legal concerns.

## REPRODUCIBILITY STATEMENT

We have made significant efforts to ensure the reproducibility of our work. Detailed descriptions of our algorithm, experimental settings, and evaluation benchmarks are provided in the main paper and appendix. To further facilitate replication, we release a subset of runnable code and scripts through an anonymous link: `https://anonymous.4open.science/status/InfoPrune`. Upon acceptance, we commit to open-sourcing the full implementation. All datasets used in our experiments are publicly available, and we carefully document processing steps to ensure transparency.

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

# APPENDIX

## A    RELATED WORK

**MLLMs**    MLLMs have witnessed rapid progress in recent years, evolving from early vision-language pretraining models such as CLIP (Radford et al., 2021) to representative systems including LLaVA (Liu et al., 2023a) (Liu et al., 2024), MiniGPT-4 (Zhu et al., 2023), and Qwen-VL (Bai et al., 2023) (Wang et al., 2024) (Bai et al., 2025). These models typically encode images into visual tokens and integrate them with textual tokens, thereby enabling strong performance on cross-modal tasks such as question answering, reasoning, and generation. Instruction tuning and large-scale multi-task

training have further advanced their capabilities, achieving breakthroughs on benchmarks such as GQA (Hudson & Manning, 2018), MMBench (Liu et al., 2023b), and VQAv2 (Goyal et al., 2016). More recent works, such as LLaVA-NeXT (Li et al., 2024) and Qwen-VL-2.5 (Bai et al., 2025), extend the frontier to long-form multimodal inputs, multi-image reasoning, and even video understanding, demonstrating stronger alignment and reasoning abilities. However, this progress comes with a substantial bottleneck: the large number of visual tokens must be propagated through all layers, which significantly increases computational overhead and inference latency. This challenge directly motivates research on visual token pruning—how to manage visual evidence efficiently while maintaining semantic completeness, thus improving the scalability and deployability of MLLMs.

**Visual Token Pruning** Existing approaches to visual token pruning can be broadly categorized into two directions: vision-centric pruning and text-centric pruning. Vision-centric methods focus on compressing or filtering visual tokens at the vision encoder or before they are fed into the LLM, aiming to remove redundancy early and reduce downstream cost. Representative examples include **FasterVLM** (Vasu et al., 2025), which reorders image tokens based on the attention between the [CLS] token and image tokens, discarding unimportant ones before entering the LLM, thereby accelerating inference; and **VisionZip** (Yang et al., 2024), which identifies dominant tokens on the vision side and aggregates redundant tokens into them. These approaches provide substantial efficiency gains but largely rely on the static assumption that "attention equals importance," which often fails to capture the true marginal contribution of tokens in semantic transmission. In contrast, text-centric pruning emphasizes cross-modal interactions inside the LLM, reducing redundant propagation of visual tokens once their contribution to textual semantics diminishes. For instance, **FastV** (Chen et al., 2024) leverages cross-attention signals to identify visual tokens that have limited influence on LLM, pruning them in deeper layers (Yin et al., 2025); **PyramidDrop** (Xing et al., 2025) adopts a layer-wise decreasing strategy, retaining more tokens at shallow layers while gradually dropping them in deeper ones, consistent with the observation that visual information is largely injected by the middle layers. While both categories alleviate computational redundancy, they generally rely on static heuristics and fail to capture the dynamic flow of information across layers. Consequently, they struggle to answer two fundamental questions: *which visual tokens truly provide new information, and when can we determine that visual information has been sufficiently absorbed by the language semantics?* Our work addresses this gap by recasting pruning as an nformation maximization problem under budget constraints, and proposes a two-stage framework based on *marginal information increment* and *semantic stability detection*, enabling efficient inference while preserving semantic completeness.

## B  IMPLEMENTATION DETAILS

For LLaVA series models, we set the $\alpha$ to 2, perform token pruning at the 16th layer ($K = 15$) of the model, and use a pruning ratio $R_2$ of 0.33. For Qwen-VL-2.5, $\alpha$ is increased to 3, and pruning is applied at the 14th layer ($K = 13$) with the same pruning ratio $R_2$ of 0.33. All experiments are conducted on a single NVIDIA A100 GPU with 80GB memory.

For the vision-side attention prior, we adopt the $-2$ layer as the reference for attention extraction. The attention scores are normalized with a temperature $\tau = 0.7$ and sharpened with exponent $\gamma_q = 0.5$. The candidate pool $\alpha$ is set differently across backbones: for the LLaVA series, $\alpha = 2$, while for Qwen-VL-2.5, $\alpha = 3$.

## C  MOTIVATION ANALYSIS

To further address the two subproblems, we present additional analyses here. Specifically, we elaborate on the design of the MFI and MGI indicators, provide extended experimental results for both, and include more results on $S_{vt}$ & $S_{vv}$.

### C.1  MFI & MGI ANALYSIS

We detail the mathematical motivations behind our MFI and MGI—from geometric and statistical perspectives. Both serve as first-order surrogates for submodular objectives that govern semantic expressiveness and statistical identifiability under token budget constraints.

### C.1.1 Geometric View: MFI as Orthogonal Residual Energy

We first interpret information as the ability of selected tokens to span novel semantic directions. Given visual embeddings $X = [x_1, \ldots, x_N]^\top \in \mathbb{R}^{N \times d}$, let $S \subset \{1, \ldots, N\}$ be a selected subset of size $t$. The natural global goal is to maximize the volume of the subspace spanned by $\mathcal{S}$:

$$\max_{|S|=T} \Phi_{\text{vol}}(S) := \log \det(G_S + \varepsilon I), \quad G_S = X_S X_S^\top.$$

This corresponds to maximizing the squared volume of the parallelotope spanned by $\{x_j\}_{j \in S}$. The log-determinant promotes diversity and additivity, making marginal analysis tractable.

Adding a candidate $x_i$ yields a marginal gain governed by Schur complement:

$$\Phi_{\text{vol}}(S \cup \{i\}) - \Phi_{\text{vol}}(S) = \log \left(1 + \|r_i\|_2^2\right),$$

where $r_i = (I - UU^\top)x_i$ is the orthogonal residual of $x_i$ onto $\text{span}(X_S)$ with $U$ being an orthonormal basis of $X_S$. We thus define the metric:

$$\boxed{\text{MFI}(x_i \mid S) := \|x_i - UU^\top x_i\|_2^2.}$$

**Geometric Interpretation.** MFI measures how far $x_i$ lies from the current span—*i.e.*, its contribution to subspace expansion. When normalized, this reduces to $\|x_i\|^2 \sin^2 \theta$, where $\theta$ is the principal angle to the current subspace. A token fully orthogonal to $X_S$ offers maximal increment; one aligned with existing directions offers none.

**Desirable Properties.**

- **Rotation-invariant**: Invariant under orthogonal transformations.
- **Monotonicity**: The residual energy decreases as $\mathcal{S}$ grows—consistent with diminishing returns.
- **Submodular alignment**: Closely approximates the greedy selection rule for maximizing $\log \det(G_S + \varepsilon I)$, a near-submodular objective with approximation guarantees.

### C.1.2 Statistical View: MGI as Fisher Information Gain

We now consider statistical identifiability. Suppose $x_j$ are drawn from an unknown distribution parameterized by latent variables, and the goal is to select tokens that best preserve global representational coverage.

Let $C_S := \frac{1}{|S|} \sum_{j \in S} x_j x_j^\top$ be the empirical covariance matrix of selected tokens. The D-optimal design criterion aims to maximize the Fisher information, leading to:

$$\max_{|S|=T} \Phi_{\text{D}}(S) := \log \det(C_S + eI).$$

This objective aligns with minimizing posterior variance or maximizing mutual information under Gaussian assumptions.

When adding token $x_i$ to $\mathcal{S}$ of size $t$, we define:

$$C_{S \cup \{i\}} = \frac{t}{t+1} C_S + \frac{1}{t+1} x_i x_i^\top.$$

Using matrix determinant lemma:

$$\Phi_{\text{D}}(S \cup \{i\}) - \Phi_{\text{D}}(S) = \log \left(1 + \tfrac{1}{t+1} x_i^\top (C_S + eI)^{-1} x_i\right).$$

We define the metric:

$$\boxed{\text{MGI}(x_i \mid S) := \log \left(1 + \tfrac{1}{t+1} x_i^\top (C_S + eI)^{-1} x_i\right).}$$

**Statistical Interpretation.** MGI quantifies how much additional variance or mutual information $x_i$ contributes, conditioned on $\mathcal{S}$. It prefers directions underrepresented in $C_S$, boosting the spectrum's smallest eigenmodes—thus improving global identifiability.

**Desirable Properties.**

- **Spectral sensitivity**: Higher gains for directions orthogonal to high-variance modes.
- **Submodularity**: D-optimal design is provably submodular, ensuring diminishing marginal returns.
- **Information-theoretic soundness**: In the Gaussian setting, MGI directly reflects mutual information increment between selected tokens and target labels.

### C.1.3 COMPLEMENTARITY AND ALIGNMENT WITH BUDGET OBJECTIVES

MFI and MGI serve complementary roles:

- **MFI** emphasizes *local geometric novelty*, ensuring tokens expand the span of semantic directions.
- **MGI** emphasizes *global statistical contribution*, preferring tokens that enrich underrepresented structures.

Jointly, they reflect a principled criterion for visual token pruning under budget constraints: avoid redundant directions and reinforce diverse, high-information signals. Their derivations connect to core theories in linear algebra, experimental design, and information theory—providing a solid theoretical foundation for information-guided selection.

### C.2 MORE MFI & MGI RESULTS

#### C.2.1 SETTINGS

In the measurement setup, for LLaVA we record statistics every 8 steps. For Qwen-VL-2.5, since the number of visual tokens varies dynamically, we record every $1/72$ of the sequence and normalize the overall trajectory to $1$.

#### C.2.2 MORE RESULTS

We conduct experiments on three models over SQA, MME, MMbench, and POPE. The results are illustrated in Fig 6 - Fig 8. Consistent with the observations in Fig 2, all three models exhibit substantial fluctuations in both indicators. This suggests that the attention-based selection strategy is not information-optimal, as it fails to ensure stability or reliability from the perspective of marginal information contribution.

#### C.2.3 MORE RESULTS FOR INFOPRUNE

We additionally report the MFI and MGI indicators under our proposed method, as shown in the Fig 10 - Fig 12. It can be observed that both metrics are consistently improved across all three model families.

### C.3 MORE $S_{vt}$ & $S_{vv}$ RESULTS

We also present additional results of $S_{vt}$ & $S_{vv}$ for other models. As shown in the Fig 13 and 14, similar to LLaVA-1.5 (Fig 3), both LLaVA-Next and Qwen-VL-2.5 exhibit a slowdown of visual information injection in the middle layers, providing empirical support for mid-layer pruning in the LLM backbone.

### C.4 SEMANTIC MATURITY RESULTS

Due to space limitations, we present here the experimental results on semantic maturity, as shown in the Fig 15.

## D ALGORITHM

We present the first stage of our algorithm, whose complete procedure is detailed in Algo 1.

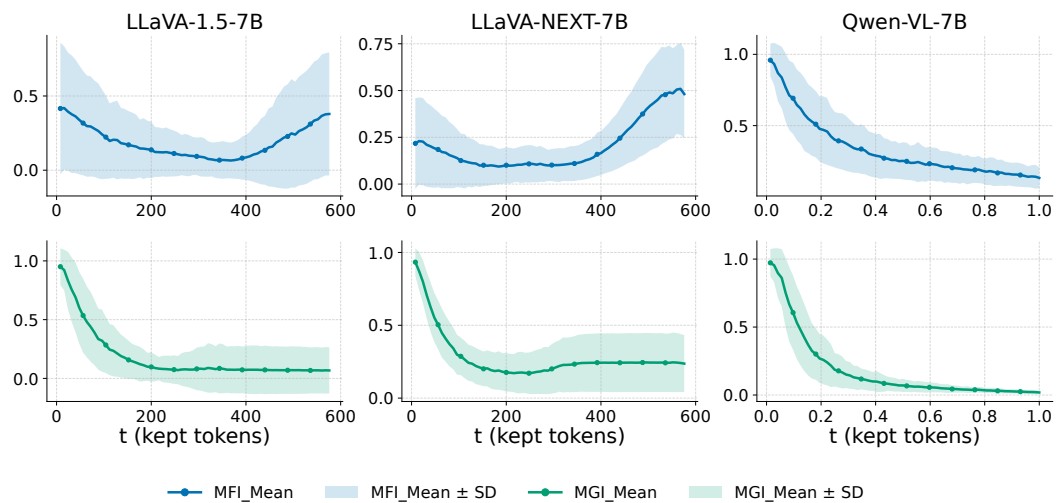

Figure 6: Visualization of MFI & MGI under the attention-based token selection strategy in on MMbench.

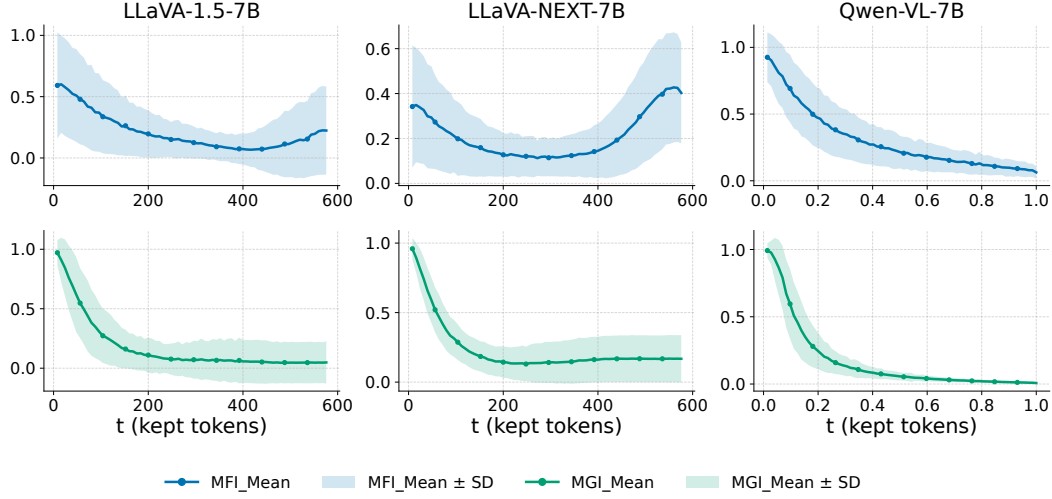

Figure 7: Visualization of MFI & MGI under the attention-based token selection strategy in on MMbench.

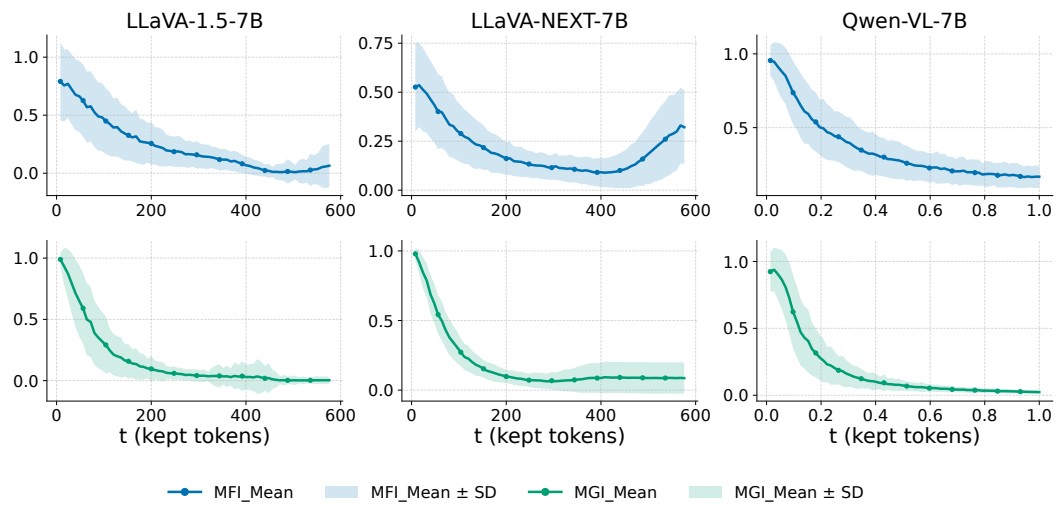

Figure 8: Visualization of MFI & MGI under the attention-based token selection strategy on POPE.

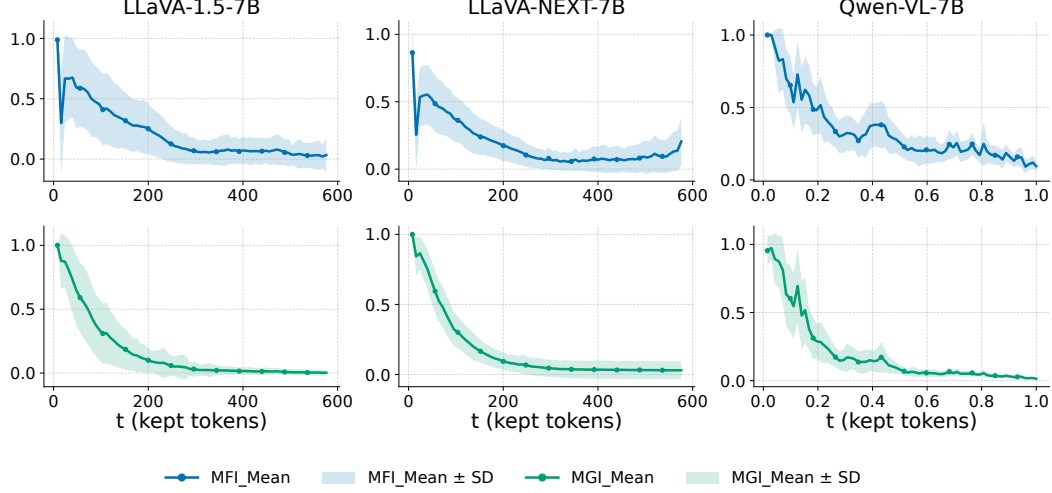

Figure 9: Visualization of MFI & MGI under the InfoPrune token selection strategy on SQA.

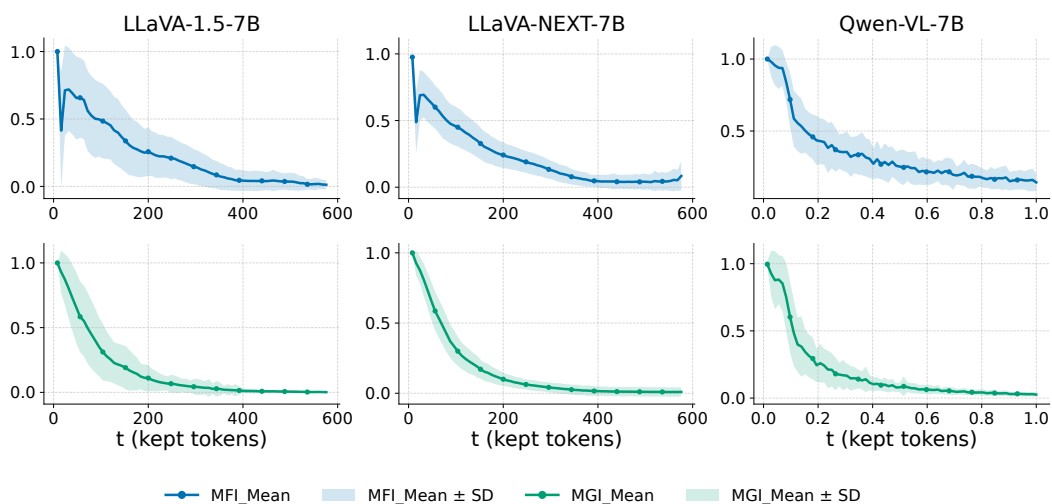

Figure 10: Visualization of MFI & MGI under the InfoPrune token selection strategy on MME.

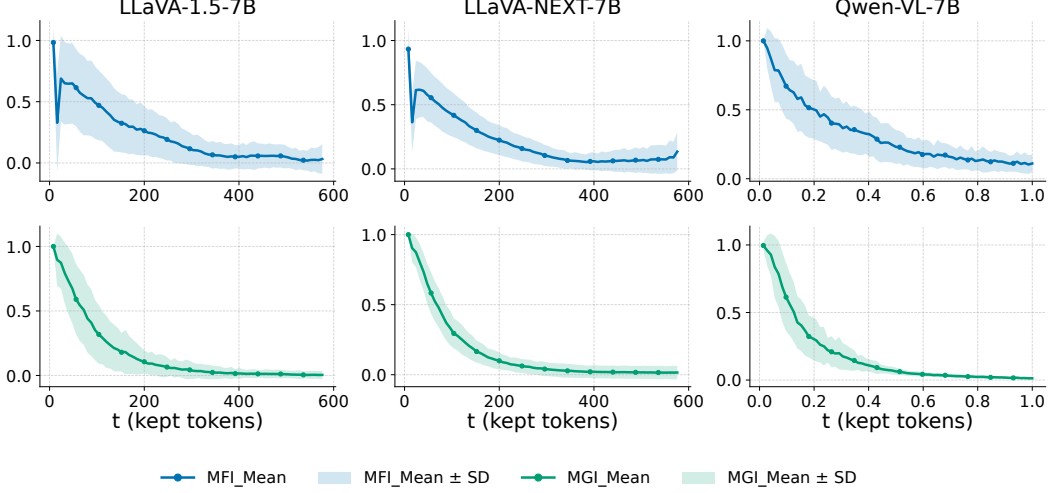

Figure 11: Visualization of MFI & MGI under the InfoPrune token selection strategy on MMbench.

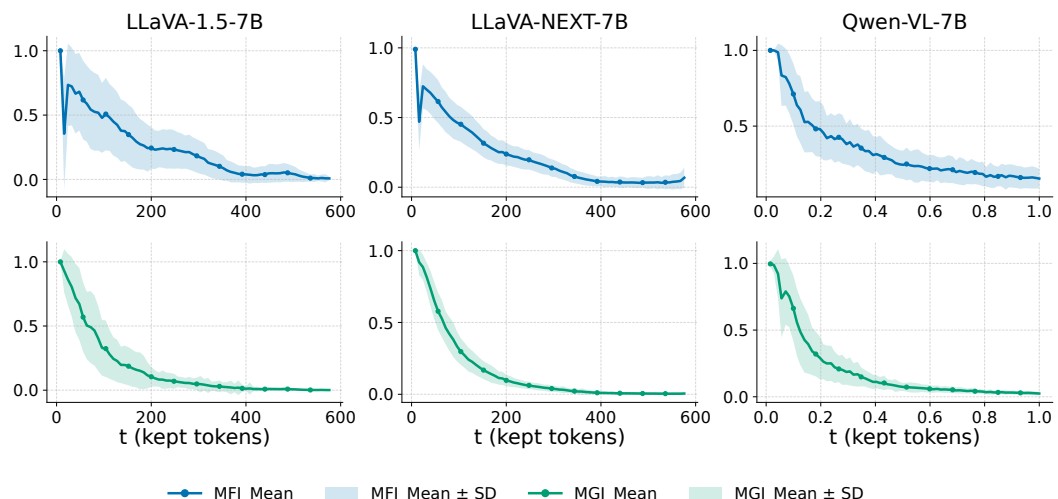

Figure 12: Visualization of MFI & MGI under the InfoPrune token selection strategy on POPE.

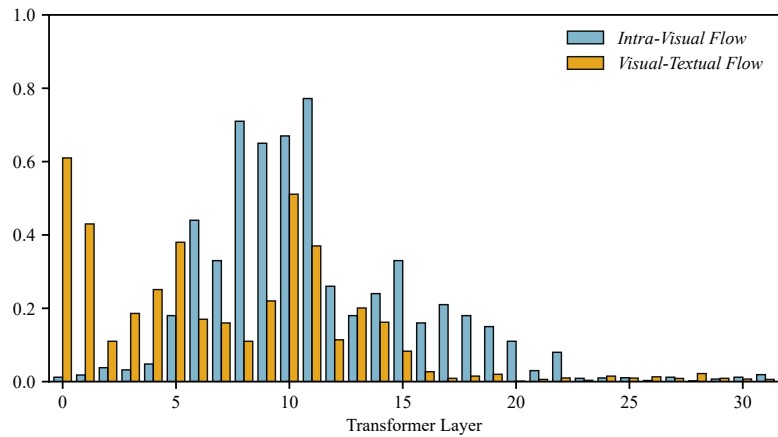

Figure 13: **Layer-wise evolution of $S_{vt}$ and $S_{vv}$ for LLaVA-NEXT-7B on POPE.** Visual information is massively injected from shallow to mid layers, after which its intensity steadily decays.

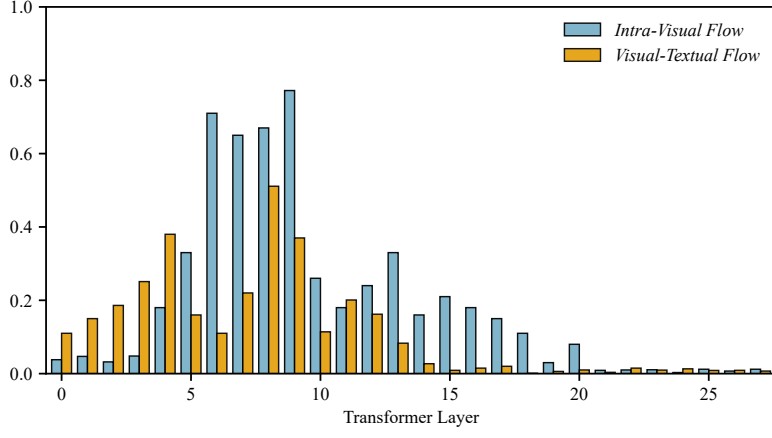

Figure 14: **Layer-wise evolution of $S_{vt}$ and $S_{vv}$ for Qwen-VL-2.5-7B on POPE.** Visual information is massively injected from shallow to mid layers, after which its intensity steadily decays.

Figure 15: **Layer-wise visualization of semantic prediction dynamics on the POPE dataset using LLaVA-1.5-7B.** Each row corresponds to a visual question instance, and each column denotes a decoder layer from 0 to 31. At each layer, we extract the model's top-1 predicted token, with background color indicating its softmax probability.

## D.1 ALGORITHMIC ANALYSIS OF STAGE I

In this subsection, we provide a theoretical justification for the design of our Stage I Algorithm 1, which performs vision-side information-increment pruning and consolidation. The algorithm combines three components—an attention-based prior, a DPP-style reweighting, and greedy residual selection—that jointly approximate the objective of maximizing semantic coverage under a strict pruning budget.

**Connection to submodular maximization.** The pruning objective can be naturally expressed as selecting a subset $S \subseteq \{1, \dots, N\}$ of size $T$ that maximizes an information functional $f(S)$, such as the log-determinant of the Gram matrix or the Fisher information of selected embeddings (Dereziński et al., 2019). Both objectives are known to be *submodular*, *i.e.*, they satisfy the diminishing returns property:

$$f(S \cup \{i\}) - f(S) \ \geq \ f(S' \cup \{i\}) - f(S'), \quad \text{whenever } S \subseteq S'.$$

For submodular maximization under a cardinality constraint, greedy selection provides a $(1 - 1/e)$-approximation guarantee. Our residual-based greedy criterion

$$r_i = \|Z_i\|_2^2 - \|U^\top Z_i\|_2^2 = \|(I - P_U)Z_i\|_2^2,$$

is precisely the first-order marginal gain in $\log \det G_S$, where $G_S$ is the Gram matrix of the current selection. This establishes that each chosen token contributes a novel (Belhadji et al., 2020), non-redundant direction to the representational subspace.

**Role of DPP reweighting.** Prior to greedy selection, we apply a determinantal point process (DPP)-style scaling $Z_i = \sqrt{q_i}\, x_i$, where $q_i$ encodes attention-based priors (Kulesza & Taskar, 2012). This operation aligns the sampling distribution with the determinant objective: high-attention tokens are given larger magnitudes, but redundancy among them is penalized due to the determinant's diversity bias. In effect, this initialization balances local saliency (via $q_i$) and global diversity (via orthogonalization), serving as a tractable approximation to DPP subset sampling.

**Greedy residual selection as Orthogonal Matching Pursuit.** The iterative update with orthogonal projection is mathematically equivalent to Orthogonal Matching Pursuit (OMP) in sparse approximation (Han et al., 2017) (Gartrell et al., 2019). At each step, the residual norm $r_i$ measures the energy of a token orthogonal to the current span. Adding the maximizer ensures monotone increase in covered subspace volume. The stopping condition $\max_i r_i \leq \varepsilon$ corresponds to novelty saturation, where additional tokens no longer provide meaningful incremental information.

**Token Merge.** After subset selection, our algorithm merges discarded tokens into their nearest prototypes. This step does not affect the theoretical subset selection, but ensures that discarded features are softly re-incorporated, stabilizing optimization without altering the $(1-1/e)$ approximation guarantee of the greedy stage.

---

**Algorithm 1** Vision-side Information-Increment Pruning and Consolidation (Attn $\to$ DPP(Linear) $\to$ Greedy)

---

**Require:** Token embeddings $X = [x_1, \ldots, x_N]^\top \in \mathbb{R}^{N \times d}$ (row-wise), deep attention scores $a \in \mathbb{R}^N$ from visual encoder's [CLS] token, pruning budget $T \ll N$ (with ratio $R_1 = T/N$), over-provision factor $\alpha > 1$, temperature $\tau_{\text{attn}}$, exponent $\gamma_q$, saturation threshold $\varepsilon > 0$, inertia $\beta > 0$.

**Ensure:** Selected index set $\mathcal{S}$ with $|\mathcal{S}| = T$, consolidated features $\tilde{X} \in \mathbb{R}^{N \times d}$ (same shape as $X$).

1: **// Attention prior and candidate pool**
2: $q_i \leftarrow \left(\text{softmax}(a/\tau_{\text{attn}})_i\right)^{\gamma_q}$ for $i = 1..N$      $\triangleright$ attention-based importance score
3: $\mathcal{C} \leftarrow \text{TopK}_{\alpha T}\left(\{q_i\}_{i=1}^N\right)$      $\triangleright$ top-$\alpha T$ candidate pool
4: $Z_i \leftarrow \sqrt{q_i}\, x_i$ for $i \in \mathcal{C}$      $\triangleright$ DPP-style scaled features

5: **// Greedy selection by residual information gain (OPP)**
6: $S \leftarrow \varnothing$;   $U \leftarrow []$      $\triangleright$ $U \in \mathbb{R}^{d \times |S|}$: orthonormal basis of $\text{span}(Z_S)$
7: **for** $t = 1$ **to** $T$ **do**
8:      **for all** $i \in \mathcal{C} \setminus S$:   $r_i \leftarrow \|Z_i\|_2^2 - \|U^\top Z_i\|_2^2$      $\triangleright$ marginal gain (residual energy)
9:      **if** $\max_{i \in \mathcal{C} \setminus S} r_i \leq \varepsilon$ **then**
10:          **break**      $\triangleright$ stop if novelty gain saturates
11:      **end if**
12:      $i^\star \leftarrow \arg\max_{i \in \mathcal{C} \setminus S} r_i$
13:      $S \leftarrow S \cup \{i^\star\}$
14:      $v \leftarrow Z_{i^\star} - U\left(U^\top Z_{i^\star}\right)$;   $u \leftarrow v/\|v\|_2$
15:      $U \leftarrow [U \mid u]$      $\triangleright$ append orthogonal direction
16: **end for**

17: **// Fill or truncate based on attention if needed**
18: **if** $|S| < T$ **then**
19:      $F \leftarrow \text{TopK}_{T-|S|}\left\{q_i : i \in \mathcal{C} \setminus S\right\}$;    $S \leftarrow S \cup F$
20: **end if**
21: **if** $|S| > T$ **then**
22:      $S \leftarrow \text{TopK}_T\left\{q_i : i \in S\right\}$      $\triangleright$ tie-breaking truncation
23: **end if**

24: **// Token merge**
25: $\tilde{X} \leftarrow X$;    $R \leftarrow \{1, \ldots, N\} \setminus S$
26: Initialize counts $c_j \leftarrow \beta$ and accumulators $\tilde{x}_j \leftarrow \beta\, x_j$ for all $j \in S$
27: **for** each $i \in R$ **do**
28:      $\hat{j}(i) \leftarrow \arg\max_{j \in S} \dfrac{\langle x_i, x_j \rangle}{\|x_i\|_2\, \|x_j\|_2}$      $\triangleright$ assign to nearest prototype
29:      $c_{\hat{j}(i)} \leftarrow c_{\hat{j}(i)} + 1$;    $\tilde{x}_{\hat{j}(i)} \leftarrow \tilde{x}_{\hat{j}(i)} + x_i$
30: **end for**
31: **for** each $j \in S$ **do**
32:      $\tilde{x}_j \leftarrow \tilde{x}_j/c_j$;    $\tilde{X}[j] \leftarrow \tilde{x}_j$      $\triangleright$ overwrite prototype embeddings
33: **end for**
34: **return** $(S, \tilde{X})$

---

**Summary.** In total, Stage I can be viewed as an efficient surrogate for solving a submodular maximization problem: attention priors supply a computationally cheap relevance signal, DPP scaling injects diversity, and greedy residual selection provides approximation-theoretic guarantees. This layered design yields a principled balance between tractability, information coverage, and computational efficiency.

| Pool $\alpha$ | 64 | 128 | 192 |
|---|---|---|---|
| 1.2 | 1691 | 1771 | 1797 |
| 1.5 | 1706 | 1775 | 1802 |
| 2 | 1715 | 1780 | 1817 |
| 2.2 | 1707 | 1763 | 1817 |
| 2.5 | **1716** | 1777 | 1817 |

| Pool $\alpha$ | 11.1 | 22.2 |
|---|---|---|
| 1.5 | 1891 | 2045 |
| 2 | 1904 | 2076 |
| 2.5 | 1915 | 2103 |
| 3 | 2004 | 2131 |
| 3.5 | 1975 | 2116 |

Table 5: Ablation on candidate pool ratio $\alpha$ for LLaVA-1.5 in MME.

Table 6: Ablation on candidate pool ratio $\alpha$ for Qwen-VL-2.5 in MME.

## E  ABLATION STUDY

**Ratio $R$.** We conduct an ablation study on the retention ratio settings for the two pruning stages under a fixed overall budget $R = 11.1\%$. As shown in Tab 4, we evaluate the performance of LLaVA-1.5 on SQA and MME under various configurations of $R_1$ and $R_2$. The results demonstrate that multiple configurations can effectively preserve model accuracy. For better trade-off and stability, we set the second-stage pruning ratio to $R_2 = 33\%$ in the main experiments.

**Candidate Pool Ratio $\alpha$.** We first ablate the candidate pool ratio $\alpha$ used in Stage 1. As shown in Tab 5 and Tab 6, we experiment with three pruning configurations on LLaVA-1.5 and evaluate their performance on MME. When $\alpha$ is small, the selection strategy closely resembles raw attention ranking, resulting in suboptimal performance retention. As $\alpha$ increases, the model benefits from broader candidate coverage and achieves higher fidelity. However, overly large $\alpha$ dilutes the prior imposed by attention, slightly reducing effectiveness. Based on overall trends, we choose $\alpha=2$ for LLaVA models and $\alpha=3$ for Qwen-VL-2.5.

Table 4: Ablation on retention configurations $R_1$ & $R_2$.

| $R_1$ | $R_2$ | SQA | MME |
|---|---|---|---|
| 11.1% | 100.0% | **69.8** | 1691 |
| 13.3% | 66.7% | 67.1 | 1674 |
| 14.8% | 50.0% | 67.5 | 1700 |
| 16.7% | 33.3% | **69.8** | 1715 |
| 22.2% | 0.0% | 68.5 | **1765** |

**Pruning Depth $K$.** We also conduct an ablation on the pruning depth $K$, which determines the layer at which visual tokens are removed. As shown in Tab 7, early pruning consistently leads to severe performance degradation. In contrast, pruning at middle layers significantly mitigates the loss—aligning with our experimental findings in the motivation section (Insight 2) on visual injection and semantic maturity. Given that LLaVA models have 32 layers and Qwen-VL-2.5 has 28, we choose $K=15$ and $K=13$ respectively to strike a balance between efficiency and performance.

| Layer $K$ | LLaVA-1.5 | LLaVA-NEXT | Qwen-VL-2.5 |
|---|---|---|---|
| 7 | 1544 | 1762 | 1746 |
| 9 | 1653 | 1758 | 1747 |
| 11 | 1695 | 1796 | 1774 |
| 13 | 1701 | 1797 | 2004 |
| 15 | **1715** | **1797** | **2109** |
| 17 | 1701 | 1797 | 2127 |
| 19 | 1700 | 1798 | 2116 |

Table 7: Ablation on pruning layer $K$. Mid-layer pruning (e.g., $K = 15$) balances semantic maturity and FLOPs reduction.

## F  QUALITATIVE ANALYSIS

Beyond quantitative evaluation, we also perform qualitative analysis on LLaVA-Bench Liu et al. (2023a). As illustrated in Fig 16 and Fig 17, InfoPrune produces only a few hallucinations in model outputs, which validates the effectiveness of our method.

What might be the intended effect of this painting?

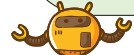
• Red-highlighted text
Indicates incorrect answers.

The intended effect of this painting is to create a humorous and whimsical scene by combining a portrait of a dog with a human-like appearance, giving it the appearance of a person. The dog is painted with a human-like face, complete with a beard, and is positioned in a way that it seems to be looking at the viewer. This artistic choice is likely meant to evoke amusement and surprise, as it is not common to see a dog portrayed in such a manner. The painting may also serve as a conversation starter or a unique piece of art that showcases the artist's creativity and sense of humor.

The intended effect of this painting is to create a humorous and whimsical scene by combining a dog with human-like features, such as a face and a neck, and placing it in a human-like setting. The dog is depicted sitting in a chair, wearing a necklace, and even has a human-like face with a nose and mouth.

Retain 11.1%          Retain 22.2%

Figure 16: **The results of LLaVA-1.5-7B on LLaVA-bench.** The method produces only a few hallucinations, while at 22.2% retention, it yields hallucination-free outputs.

What might be the intended effect of this painting?

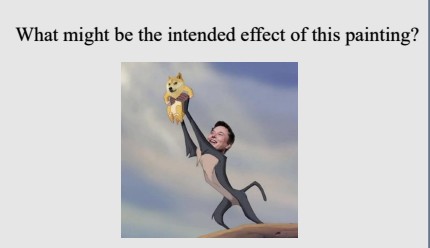
• Red-highlighted text
Indicates incorrect answers.

The meme is parodying the iconic movie scene from \"The Lion King\" where a young Simba is being held by his father, Mufasa. In the meme, a man is holding a stuffed animal, specifically a teddy bear, instead of a young lion cub. The man is also smiling, which adds a humorous twist to the scene. This parody creates a lighthearted and entertaining moment by replacing the serious and emotional context of the original scene with a playful and comical one.

The meme is parodying the iconic movie scene from \"The Lion King\" where a young Simba is being held up by his father, Mufasa. In the meme, a man is holding a baby lion cub, which is a playful and humorous twist on the original scene. The man is also smiling, adding to the lighthearted nature of the parody.

Retain 11.1%          Retain 22.2%

Figure 17: **The results of LLaVA-1.5-7B on LLaVA-bench.** The method produces only a few hallucinations when 22.2% retention.

# G  BENCHMARK

We provide a detailed description of 9 benchmarks here.

GQA (Hudson & Manning, 2018): A benchmark with 22 million compositional visual questions derived from scene graphs from Visual Genome. Questions cover attributes, relationships, and multi-step inference. Answers are restricted to a fixed vocabulary to reduce open-ended noise. Enables evaluation of reasoning and relational understanding.

MMBench (Liu et al., 2023b): A bilingual multiple-choice benchmark evaluating vision-language models across many fine-grained abilities (OCR, counting, commonsense, etc.). Incorporates CircularEval and mechanisms to map open responses into fixed choices, reducing evaluation noise. Includes versions in English and Chinese.

POPE (Li et al., 2023): Polling-based object probing evaluation for detecting object hallucination. Models are asked binary presence/absence questions about objects with annotated bounding boxes, to measure false positives when predicting non-existent objects.

MME (Fu et al., 2023): Vision-language evaluation dataset containing image-question pairs used in the LMMs-Eval (LMMS-Eval) suite. Used to test fundamental understanding abilities. (Specific task count and split details need internal confirmation.)

MMBCN (Li et al., 2022): A large Chinese multimodal benchmark with 120,000 image-text pairs and 60,000 QA pairs. Includes domain-specific scenarios (street signage, classical text, dialects), with 35% OCR content, crowd sourcing + expert validation. (Based on your internal description.)

SQAIMG (Huang et al., 2023): Contains 25,000 camera-shot images and GAN/ diffusion generated counterparts, each annotated with multiple human judgments of image quality (realism, artifacts, semantics, aesthetic). Also includes MOS scores and generation metadata. (Some details from internal source.)

VizWiz (Gurari et al., 2018): Real-world images taken by visually impaired users ( 31,000), with 50,000 spoken questions. Images include realistic noise; questions are practical. Dataset includes splits for train/val/test, crowd answers plus confidence annotations.

TextVQA (Singh et al., 2019): Benchmarks where correct answers require understanding text present in images: signs, menus, documents etc. Open vocabulary OCR, and ground truth text-visual alignment provided.

VQA-v2 (Goyal et al., 2016): Over one million QA pairs on 200,000 images. Paired images per question to reduce answer bias. Multiple annotators per question; filter for consistency. Widely used general VQA benchmark.

# H  LIMITATIONS AND DISCUSSION

While InfoPrune demonstrates strong performance retention and efficiency gains across multiple backbones and benchmarks, several limitations remain. First, our pruning framework is designed to be training-free and model-agnostic, but this comes at the cost of not exploiting task-specific signals. In highly specialized domains (e.g., OCR-intensive tasks or fine-grained recognition), the generic information metrics (MFI and MGI) may overlook subtle yet semantically critical tokens. Second, Stage 2 relies on detecting semantic convergence at a predefined mid-layer $K$, which, although empirically robust, may vary across architectures and tasks. An adaptive mechanism for automatically locating the saturation point could further improve generality. Third, while our analysis focuses on token-level information flow, other efficiency bottlenecks such as long-sequence text processing or cross-modal memory usage are not directly addressed. Finally, InfoPrune operates purely at inference time; integrating pruning signals into training might enable models to better internalize information budgeting, potentially unlocking further efficiency–accuracy trade-offs.

Despite these limitations, we believe InfoPrune offers a principled starting point for rethinking visual token management, and future work can extend our framework toward adaptive, task-aware, and training-integrated pruning strategies.

## I  THE USE OF LARGE LANGUAGE MODELS (LLMS)

In preparing this manuscript, large language models (LLMs) were used solely for minor grammar refinement and language polishing. LLMs did not contribute to research ideation, experimental design, data analysis, or result interpretation. The authors take full responsibility for the content of this paper.

