# OpenReview forum: "InfoPrune: Revisiting Visual Token Pruning from an Information-Theoretic Perspective"
_ICLR.cc/2026/Conference — ICLR 2026 Conference Withdrawn Submission_

### Official Review · Reviewer_tyby · 2025-10-25

**Soundness:** 2
**Presentation:** 3
**Contribution:** 3
**Rating:** 6
**Confidence:** 3

**Summary:**

This paper addresses the high inference overhead of MLLMs caused by dense, unselected visual token propagation. It redefines visual token pruning as an information maximization problem under budget constraints (i.e., selecting tokens with high marginal information and stopping propagation once their information is fully injected). To solve this, the authors propose InfoPrune, a training-free two-stage framework: Vision-side Pruning and LLM side pruning. The novel parts lie in the vision-side pruning, where it combines attention priors with two new metrics, Marginal Feature Increment (MFI, for geometric novelty) and Marginal Gain of Information (MGI, for statistical diversity). It is able to select non-redundant visual tokens via a Determinantal Point Process (DPP) and token merge.

Experiments on three mainstream MLLMs (LLaVA-1.5, LLaVA-Next, Qwen-VL-2.5) and 9 benchmarks show strong results: retain ~97% performance with only 11.1% visual tokens, outperforming baselines like FastV and VisionZip in generality, stability, and efficiency.

**Strengths:**

- This paper’s information-theoretic formulation is a key contribution that goes beyond only attention scores across layers.
- The method seems rigorous with theoretical grounding: MFI and MGI are linked to submodular optimization (ensuring approximation guarantees) and information theory (e.g., MGI connects to Fisher information).
- The coverage of experiments is comprehensive, though such a suite of evaluation has already been a standard for visual token reduction.
- The empirical validation shows strong performance.

**Weaknesses:**

- the performance doesn't seem to be state-of-the-art. Check out [1], and the performance is of a similar range.
- Though the whole idea of visual token reduction is to improve efficiency, there is not a run time analysis to show that such a method is actually faster. Besides, no evidence that this method is supported under FlashAttention, which further limits the actual usage of the method.

[1] Zhang et al. VScan: Rethinking Visual Token Reduction for Efficient Large Vision-Language Models

**Questions:**

See weaknesses.

---

### Official Review · Reviewer_im4J · 2025-10-30

**Soundness:** 3
**Presentation:** 2
**Contribution:** 2
**Rating:** 4
**Confidence:** 5

**Summary:**

InfoPrune introduces a two-stage, training-free pruning framework for multimodal large-language models (MLLMs).
Stage 1 performs information-increment-based visual-token selection via Marginal Feature Increment (MFI) and Marginal Gain of Information (MGI) metrics combined with DPP-style diversity weighting.
Stage 2 detects mid-layer semantic convergence to stop visual-token propagation once information flow saturates.
The method yields over 96 % performance retention with ≈ 11 % of tokens on LLaVA-1.5, LLaVA-Next, and Qwen-VL-2.5, outperforming FastV, PyramidDrop, VisionZip, and SparseVLM while delivering 1.3–1.5 × inference speed-ups.

**Strengths:**

1. Reframes pruning as an information optimization problem, connecting geometric and statistical views.
2. Works across MLLMs without retraining while achieving relatively strong FLOP reduction.

**Weaknesses:**

1. Missing an important related work: DivPrune (CVPR 2025) is not cited or compared, despite clear methodological overlap.
2. The mid-layer pruning point is manually tuned rather than automatically determined, which makes robustness a concern of the paper.
3. Beyond semantic-stopping, the contributions extend existing ideas rather than introducing new paradigms.
4. The cost of DPP-based selection is not analyzed relative to simpler pruning methods. The method runtime overhead should be clearly stated.
5. Poor presentation quality: Duplicated text (even in the abstract) and grammatical issues detract from readability.

**Questions:**

1. How does your DPP-based selection differ in formulation or complexity from DivPrune?
2. Can you include a direct comparison with DivPrune on shared benchmarks?
3. Is the mid-layer semantic convergence layer K adaptively determined or fixed?
4. What is the runtime overhead introduced by Stage 1?
5. Could the proposed metrics guide training-time sparsity?

---

### Official Review · Reviewer_vuZQ · 2025-10-30

**Soundness:** 3
**Presentation:** 2
**Contribution:** 2
**Rating:** 2
**Confidence:** 5

**Summary:**

This paper proposes InfoPrune, a two-stage training-free visual token pruning method for multimodal large language models (MLLMs). The first stage selects visual tokens based on a marginal feature/information gain criterion, aiming to retain tokens that provide unique and useful information. The second stage prunes remaining tokens during inference when semantic saturation is detected in the language model. The authors justify their method from an information-theoretic perspective and empirically demonstrate its effectiveness across several MLLMs (e.g., LLaVA-1.5, Qwen-VL-2.5), showing strong performance with significantly reduced token counts. However, there are still multiple significant issues remaining.

**Strengths:**

(S1) The method is training-free and aims to improve inference efficiency of MLLMs by selecting and truncating visual tokens in a theoretically motivated manner. The general framework—first selecting informative tokens, then stopping unnecessary token flow—is reasonable and aligns with current concerns in efficient large model inference.

(S2) The empirical results show that InfoPrune maintains high task performance (around 96–97%) while pruning over 85% of visual tokens, across multiple MLLMs. The results are competitive and demonstrate some robustness.

**Weaknesses:**

(W1)The paper claims that prior methods assume that important tokens can be easily identified via static features. However, where exactly is this “easily” stated or assumed in previous work? The paper further states that “these methods often rely on a simplified assumption that token importance can be directly inferred from local features or static scores (e.g., attention weights).” But why are attention weights considered static scores or local features? The methods compared in Figure 1 are not static at all; they are sample-wise and assign different importance scores depending on the input. Moreover, whether pruning is done in the vision encoder stage (e.g., VisionZip, which usually selects tokens after the last encoder layer) or in the LLM, there is always feature aggregation involved, so the features being used are not local. I do not understand the basis for the paper’s description of prior work in this way, and I believe this characterization is not rigorous.

(W2)The statement “most existing approaches approximate token importance via attention scores assigned by the [CLS] token, under the assumption that higher attention implies higher utility” is also not rigorous, and in fact incorrect. As shown in Figure 1, only VisionZip uses CLS token attention. Other methods do not use CLS attention at all. Also, some vision encoders in MLLMs (e.g., Qwen2.5-VL) do not even have a CLS token, so there is no such CLS attention score to begin with.

(W3)Regarding the so-called “marginal information increment” and its relation to global semantic information: how do Equation (1) and Equation (2) theoretically prove that these scores directly measure model performance or correlate with actual performance? The text uses statements like “if information sufficiency is understood as covering the principal directions of the full set…” and “if information sufficiency is viewed in terms of maximizing parameter identifiability under a linear–Gaussian proxy…”. Since the paper uses “if” as a premise, when does this hold and when does it not? Under what conditions are these assumptions mathematically valid?

(W4)An information-theoretic visual token compression method for MLLMs was already proposed earlier in TokenCarve: Information-Preserving Visual Token Compression in Multimodal Large Language Models (2025). Therefore, the information-theoretic perspective of this paper is not new. In addition, this paper does not provide conceptual distinction or experimental comparison with TokenCarve. As far as I know, TokenCarve does not rely on attention for token selection and can still achieve strong performance.

(W5)I cannot understand the conclusion of “Core Insight I.” The fact that the distributions of MFI and MGI differ across timesteps and samples actually indicates that attention is dynamic. Shouldn’t different tasks and samples naturally require different token selections? Moreover, even if attention scores are inconsistent with MFI and MGI, what does that prove? MFI and MGI appear to be measures of information contribution, but how is it shown that they correlate with model performance?

(W6)In Figure 2 (Qwen-VL-7B), why is the x-axis (kept tokens) shown as a decimal number?

(W7)The metric in Equation (4) is interesting, but it is unclear why it involves a loss function. The method is claimed to be training-free, so how is this loss obtained during inference? How is the model computing this during inference? It appears that this metric is evaluated using a dataset (e.g., POPE), so is this a dataset-level conclusion or based on specific samples? Moreover, the paper criticizes the use of attention earlier, yet Equation (4) again uses attention to measure semantic saturation. Why not apply the proposed information-theoretic method here?
Overall, the metric itself is intuitive and aligns with similar findings in MLLM visual token compression literature.

(W8)The experiments show the model can maintain performance even after token reduction, but the actual acceleration is limited. Reducing visual tokens to 64 only yields a speedup of about 1.31×, which does not match the 1.5× reported in the Efficiency Study section. Does this suggest that the cost of the pruning method is high? Furthermore, the speedup measurement is not clearly defined — is this end-to-end speedup, prefill speedup, or decoding speedup?

(W9)I do not insist that the authors must test extremely large MLLMs, but the observations presented in the paper are indeed closely tied to model size. Figure 3 uses a 32-layer LLaVA-1.5-7B model. However, MLLMs differ significantly in layer depth according to their scale: for example, Qwen2.5-VL 3B, 7B, and 72B models have 36, 28, and 80 layers respectively. Therefore, to support the conclusion about semantic saturation behavior, experiments on larger-scale models are necessary.

**Questions:**

Please see weaknesses above

---

### Official Review · Reviewer_S7kD · 2025-10-31

**Soundness:** 3
**Presentation:** 3
**Contribution:** 3
**Rating:** 4
**Confidence:** 3

**Summary:**

This paper proposes InfoPrune, an information-theoretic visual token pruning framework for multimodal large language models (MLLMs). The core idea is to recast token pruning as an information maximization problem under computational constraints, focusing on two complementary questions: (1) which tokens contribute genuine marginal information, and (2) when visual evidence injection should be stopped. InfoPrune introduces a two-stage training-free pipeline—the first stage leverages attention priors with marginal information gain for token selection, while the second detects semantic convergence in the LLM to perform one-shot mid-layer pruning. Experiments on LLaVA-1.5, LLaVA-Next, and Qwen-VL-2.5 show that InfoPrune achieves over 96% performance retention with only 11.1% of visual tokens, outperforming prior pruning methods.

**Strengths:**

1. Provides a principled and interpretable perspective on visual token pruning through an information-theoretic lens.
2. The two-stage pruning design aligns well with empirical observations of visual information saturation.
3. Demonstrates solid performance retention and cross-model generalization (LLaVA and Qwen series).
4. The method is training-free and simple to integrate, making it practical for deployment.

**Weaknesses:**

1. lacks comparisons with recent or more diverse pruning and compression methods [1-4].
2. It’s unclear how much each component (e.g., DPP selection, semantic detection) contributes individually.
3. Much of the pipeline builds upon established heuristics under a new theoretical framing.
4. Focuses mainly on accuracy retention; other efficiency metrics (e.g., real-world latency, throughput on larger images) are underexplored.

[1] Bolya, Daniel, et al. "Token merging: Your vit but faster." arXiv preprint arXiv:2210.09461 (2022).
[2] Ye, Weihao, et al. "Fit and prune: Fast and training-free visual token pruning for multi-modal large language models." Proceedings of the AAAI Conference on Artificial Intelligence. Vol. 39. No. 21. 2025.
[3] Wen, Zichen, et al. "Stop looking for important tokens in multimodal language models: Duplication matters more." arXiv preprint arXiv:2502.11494 (2025).
[4] Zhong, Yiwu, et al. "Aim: Adaptive inference of multi-modal llms via token merging and pruning." Proceedings of the IEEE/CVF International Conference on Computer Vision. 2025.

**Questions:**

1. How sensitive is InfoPrune to hyperparameters like the DPP threshold, α, and pruning ratio R₂?
2. Can InfoPrune adapt dynamically to different inputs rather than using a fixed pruning layer K?
3. Have the authors considered integrating their method with recent adaptive computation or early-exit strategies for more general applicability?
4. Are there cases where information-theoretic selection fails due to noise or redundancy in visual embeddings?
5. To better validate the actual effectiveness of the proposed method, it is recommended to perform evaluation experiments within this unified framework [1]

[1] Liao, Chenfei, et al. "Are We Using the Right Benchmark: An Evaluation Framework for Visual Token Compression Methods." arXiv preprint arXiv:2510.07143 (2025).

---

### Note · Authors · 2025-11-26

I have read and agree with the venue's withdrawal policy on behalf of myself and my co-authors.